# Large-scale groundwater flow and sedimentary diagenesis in continental shelves influence marine chemical budgets

Alicia M. Wilson [1] ✉, Andrew Osborne[1,2] & Scott M. White [1]

The major ion chemistry of the ocean has been assumed to be controlled by river input, hydrothermal circulation at mid-ocean ridges, carbonate production, and low-temperature alteration of seafloor basalt, but marine chemical budgets remain difficult to balance. Here we propose that large-scale groundwater flow and diagenetic reactions in continental shelf sediments have been overlooked as an important contributor to major ion budgets in the ocean. Based on data synthesized from 17 passive margin basins, continental shelves contribute fluid exchanges comparable to hydrothermal circulation at mid-ocean ridges. Chemical exchange is similarly significant, indicating removal of $Mg^{2+}$ from the oceans at rates similar to mid-ocean ridge convection. Continental shelves likely contribute $Ca^{2+}$ and $K^+$ to the oceans at rates that, in combination with low-temperature basalt alteration, can close current budget deficits. Flow and reaction in continental shelf sediments should be included in a new generation of studies addressing marine isotope budgets.

The major ion budgets of the ocean influence marine ecosystems[1], global carbon cycling[2], and the evolution of seawater chemistry over geologic time[3,4]. In current conceptual models, the major ion chemical budgets of the ocean are controlled by river discharge, MOR hydrothermal convection, low-temperature alteration of oceanic crust on MOR flanks, and accumulation of calcium carbonate in marine environments[2,5–9] (Fig. 1). However, significant imbalances persist in these models that suggest a missing budget component. Global riverine contributions are reasonably well constrained and contribute cations ($Ca^{2+}$, $Mg^{2+}$, $Na^+$, and $K^+$) from weathered continental material. Carbonate accumulation removes ~3 Gt of carbonate per year[2,10], strongly influencing $Ca^{2+}$ and carbon budgets and also removing a small amount of $Mg^{2+}$. High-temperature MOR circulation removes $Mg^{2+}$ and contributes $Ca^{2+}$ and $K^+$ to the oceans, and a general consensus has been reached concerning the volumetric flux and fluid compositions that support this exchange[8]. Low-temperature alteration of oceanic crust removes $Mg^{2+}$ and $K^+$ and releases $Ca^{2+}$ to seawater, but the degree of alteration is highly temperature-dependent, and consequently, the rates of mass exchange are highly uncertain[11].

Current budget discrepancies include an insufficient supply of $Ca^{2+}$ to balance removal by carbonate precipitation[2], and insufficient sinks of $K^+$ and $Mg^{2+}$ to balance inputs from rivers[11,12]. A new generation of studies based on stable isotopes of Ca, K, and Mg promise significant advances in quantifying these budgets[9,13–17] and have already begun to identify new discrepancies, including the need for a major sink of $Mg^{2+}$ in the form of dolomite[15]. This rules out low-temperature basalt alteration as a way to close Mg budgets and suggests that other major chemical exchanges are required.

Suggestions that marine chemical budgets can be influenced by groundwater flow and sedimentary diagenesis in continental shelves date back decades[2,18,19] but are not included in current marine chemical budget models. Groundwater migration is widespread throughout deep sedimentary basins in continental margins[19–26], and water compositions in such settings are pervasively altered by sedimentary diagenetic reactions[27,28]. The net effect of common sedimentary diagenetic reactions like dolomitization, albitization, and illitization is to sequester $Mg^{2+}$ and $K^+$ while releasing $Ca^{2+}$ (Table 1). Note that the groundwater that participates in sedimentary diagenesis in continental shelves is overwhelmingly saline (near seawater salinity or higher,

[1]School of the Earth, Ocean and Environment University of South Carolina, Columbia, SC 29208, USA. [2]Now at: INTERA, 3 Sugar Creek Center Blvd., Suite 675, Sugar Land, TX 77478, USA. ✉e-mail: awilson@seoe.sc.edu

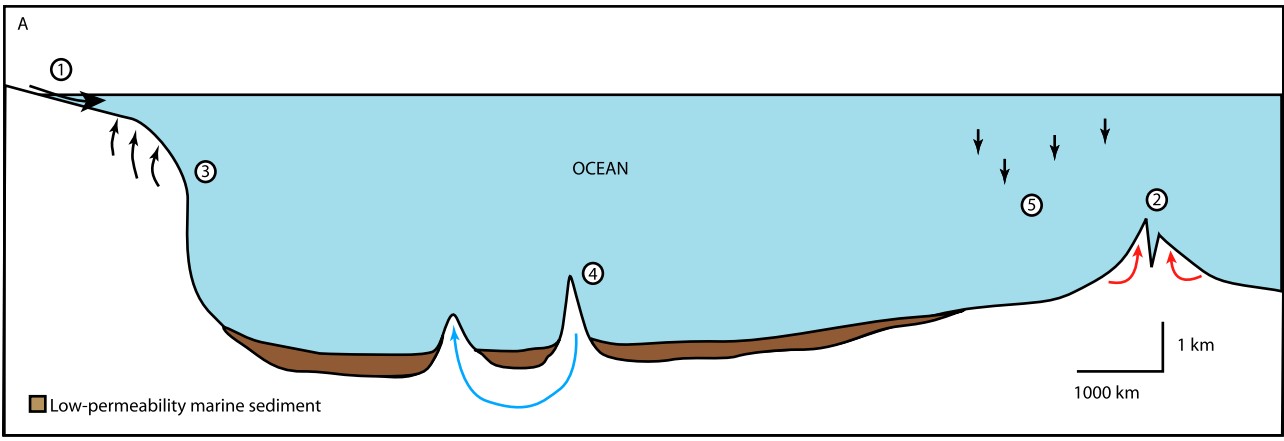

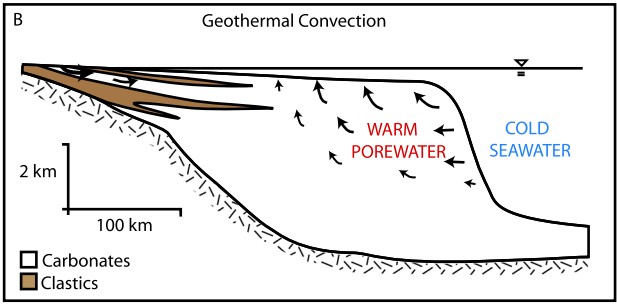
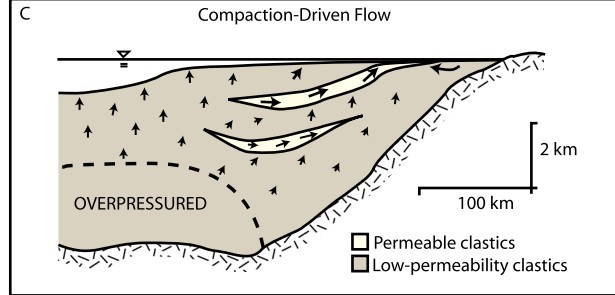

**Fig. 1 | Conceptual models. A** Processes that control the major ion chemistry of the ocean, including (1) river input, (2) hydrothermal convection at the mid-ocean ridge, (3) long-term flow and sedimentary diagenesis in passive continental margins (see **B** and **C**), (4) low-temperature hydrothermal convection and basalt alteration, and (5) carbonate precipitation. **B** Geothermal convection of seawater through passive continental margins. **C** Compaction-driven discharge of ancient porewater from a passive continental margin. (**C** redrawn by the authors of the manuscript after Garven, 1995[15]).

including brines), originating as seawater rather than rainwater. Although fresh submarine groundwater discharge (SGD) may influence marine isotope budgets[29], relatively short fluid residence times and low temperatures mean that neither fresh SGD nor tidally-driven saline SGD are likely to participate in significant sedimentary diagenetic reactions. Overall, continental margin basins have the potential to generate significant saline discharge with ion concentrations that differ markedly from seawater.

Here, we show that regional groundwater flow through continental shelves is likely a major contributor to the major ion chemistry of the ocean. We focus on flow through passive continental margins for this first-order assessment of fluid and chemical mass fluxes through continental shelves, because broad passive margins host the majority of continental shelf sediments in the world by area and volume, and the global length of passive margins is roughly twice that of convergent margins[30]. Our calculations confirm that fluxes from passive margins greatly exceed reported fluxes from compaction-driven dewatering of accretionary prisms in active continental margins[31–33]. We find that fluid and geochemical fluxes from passive margin basins are comparable to fluxes from high-temperature fluid circulation at MORs. Including chemical fluxes from passive margins and low-temperature basalt alteration allows us to close deficits in existing chemical budgets for modern seawater.

## Results and discussion
### Volumetric fluxes
In passive continental margins, flow of saline groundwater is driven by sediment compaction and fluid density gradients[34]. Global compaction-driven discharge was estimated based on the decompacted thickness and age of six representative overpressured basins (Fig. 2, Table 2). These shale-rich basins (Supplementary Fig. 1) represent ~20% of global overpressured basins, so we multiplied our result by five to obtain a global estimate. The resulting global volumetric groundwater flux from compacting sedimentary basins was $6.8 \times 10^8 \, m^3 \, yr^{-1}$, with a range of $2.7 \times 10^8 \, m^3 \, yr^{-1}$ to $1.7 \times 10^9 \, m^3 \, yr^{-1}$, accounting for uncertainty in shale content (Supplementary Fig. 2). The fluxes for individual basins (Fig. 3) varied according to their offshore area, age, and the percent shale component of the sedimentary prism. For example, the area of the GoM is slightly smaller than that of the North Sea (Table 2) yet has a much higher estimated volumetric flux due to its higher shale content. The Nam Con Son Basin has a relatively low shale component but has been actively compacting throughout its short (Table 2) history.

With respect to flow driven by density gradients in passive margins, we focused on thermal gradients rather than salinity gradients, because thermal gradients exist as a reasonably stable and predictable driver in all continental shelves[19]. Global discharge from geothermal convection was estimated based on numerical models[19] and the dominant sediment type in 11 representative basins (Fig. 2, Table 3), again representing ~20% of continental shelves globally that are likely to host geothermal convection. The resulting global volumetric groundwater flux from geothermal convection was $1.4 \times 10^{10} \, m^3 \, yr^{-1}$, with a range of $2.8 \times 10^9$ to $6.9 \times 10^{10} \, m^3 \, yr^{-1}$ when a factor of five uncertainty

## Table 1 | Examples of common water-rock reactions that impact the chemistry of groundwater

| Reaction | Formula |
|---|---|
| Dolomitization | $2CaCO_3 + Mg^{2+} \rightarrow CaMg(CO_3)_2 + Ca^{2+}$ |
| Albitization | $CaAl_2Si_2O_8 + 4SiO_2 + 2Na^+ \rightarrow 2NaAlSi_3O_8 + Ca^{2+}$ |
| Illitization[a] | Smectite + $K^+$ → Illite + $SiO_2$ + $Na^+$ + $Ca^{2+}$ + $Fe^{2+}$ + $Mg^{2+}$ + $OH^-$ + $nH_2O$ |

[a]Equation is not balanced because the stoichiometric coefficients for all constituents depend on highly variable compositions of smectite and illite.

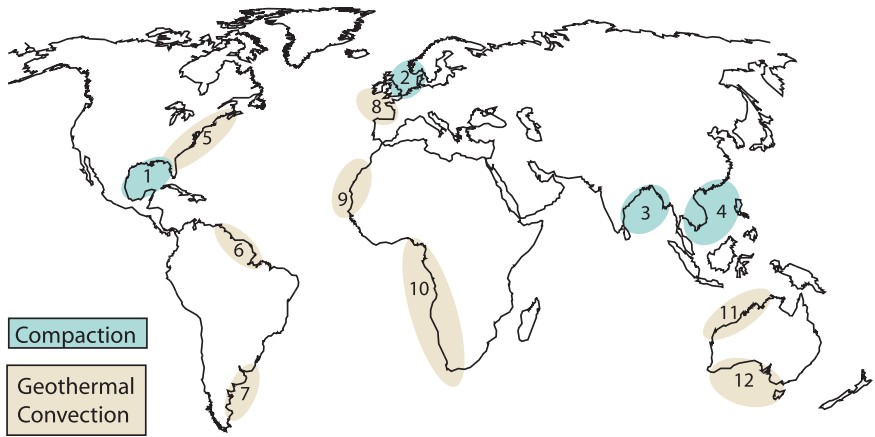

**Fig. 2 | Areas of study.** 1 – The Gulf of Mexico, 2 –North Sea, 3 – Bengal Basin, 4* – Yinggehai Basin, Nam Con Son Basin, Cuu Long Basin, 5* – North American East Coast, 6,7 – South American East Coast, 8 – Celtic Sea, 9,10 – African West Coast, 11 – Great Australian Bight, 12 – Northwest Australian shelf. *Basins were broken into separate sections and analyzed independently from each other. The continent outlines were created using GMT-6.4.0 (http://gmt.soest.hawaii.edu/projects/gmt/wiki/Download) and the GMT GSHHG coastline dataset.

related to sediment composition was considered. Estimates for volumetric discharge due to geothermal convection in individual basins ranged from $3.6 \times 10^7$ m$^3$ yr$^{-1}$ to $7.2 \times 10^8$ m$^3$ yr$^{-1}$ (Fig. 4).

Summing the global compaction-driven and geothermal fluxes yields a global volumetric flux ($Q_s$) of $1.5 \times 10^{10}$ m$^3$ yr$^{-1}$, with a range of uncertainty from $3.0 \times 10^9$ to $7.1 \times 10^{10}$ m$^3$ yr$^{-1}$. This estimate is dominated by geothermal convection. Although the fluxes per kilometer of coastline (or of basin length, for most overpressured basins) are of the same order of magnitude for geothermal convection and sediment compaction, compaction-driven flow comprises just 5% of the global flow. This is largely because compaction-driven flow is relatively uncommon in continental shelves. Note that our lower-bound estimate of global discharge from continental shelves is very comparable to the estimated volumetric flux from high-temperature MOR convection, $6.54 \times 10^9$ m$^3$ yr$^{-1}$, which we note is also subject to uncertainty of roughly a factor of four[35]. Thus, our best estimate of passive margin flux is similar to the upper-bound estimate for high-temperature MOR fluxes, and our upper-bound estimate exceeds the best estimate for high-temperature MOR convection by a factor of 25. Our estimate of global groundwater fluxes through passive margins is three orders of magnitude below estimated river discharge, $3.75 \times 10^{13}$ m$^3$ yr$^{-1}$ [6].

**Chemical mass fluxes**

We considered four example fluid compositions to estimate possible chemical mass fluxes, as a proof of concept and to inform later mass balance calculations. The four examples, hereafter referred to as archetypes, are not endmembers but rather representative of the continuum of groundwater/brine compositions encountered in deep sedimentary basin settings[36] (Supplementary Fig. 3), including representative compositions from (1) observed saline cold seeps, (2) clastic-dominated basins, (3) deep $CaCl_2$ brines[28], and (4) carbonate-dominated basins (Table 4). The cold seep archetype comes from observed brine seeps in the Gulf of Mexico. These relatively highly-reacted fluids differ somewhat from cold seep fluids in other settings, including gas hydrate systems[37,38] and convergent margins[33], but a key feature of all of these seeps is that they are depleted in $Ca^{2+}$ relative to seawater. Deep $CaCl_2$ brines represent highly evolved fluids characterized by high temperatures (>100 °C), high salinities, and long reaction times (millions of years). The clastic- and carbonate-dominated archetypes represent specific sediment compositions, in which dolomitization is excluded (clastic) or is the dominant form of diagenesis (carbonate). The sources and justifications for these archetypes are discussed in more detail in the methods.

The mass flux calculations revealed groundwater chemical fluxes that rival MOR contributions for most archetypes (Fig. 5a), indicating that groundwater contributions have the potential to influence the major ion chemistry of seawater. The pattern of inputs and outputs varies for the different archetypes and ions, with the exception of $Mg^{2+}$, which shows net removal of $Mg^{2+}$ from the ocean for every archetype. Trends are difficult to assign for $Na^+$ and $K^+$, but all archetypes except the cold seep archetype indicated the export of $Ca^{2+}$ to the ocean. The clastic and $CaCl_2$ brines show similar trends in inputs and outputs for all ions except for $K^+$, but to better interpret the pattern of inputs and outputs for the archetypes it is helpful to place our calculated fluxes in the context of the global budget deficits.

We calculated the theoretical composition of groundwater needed to balance marine chemical budgets using our calculated volumetric fluxes ($Q_s$) and assuming no contribution from low-temperature basalt alteration. To do this, we rewrote the mass balance model of Spencer and Hardie[6] to incorporate more recent estimates of volumetric fluxes from high-temperature MOR convection[35] and mass flux estimates of $CaCO_3$ accumulation[2,10]. We solved the theoretical composition of groundwater, $C_{th}$, according to

$$C_{th} = -\frac{Q_{mor}C_{mor} + Q_r C_r + M_c}{Q_s} \tag{1}$$

where $Q_{mor}$ is the volumetric flux of hydrothermal fluid through MORs, $6.54 \times 10^9$ m$^3$ yr$^{-1}$ [35], $C_{mor}$ is the adjusted concentration of hydrothermal fluids[6] (Table 4), $Q_r$ is the global volumetric river discharge, $3.75 \times 10^{13}$ m$^3$ yr$^{-1}$ (ref. 6), $C_r$ is the adjusted concentration of river water[6] (Table 4), $M_C$ is ion removal by carbonate precipitation, $-1.06 \times 10^{12}$ kg yr$^{-1}$ $Ca^{2+}$ and $-7.15 \times 10^{10}$ kg yr$^{-1}$ $Mg^{2+}$ [2], and $Q_s$ is the volumetric flux of groundwater from passive margins. This allowed us

**Table 2 | Basins likely to host compaction-driven flow**

| Basin | Age (My) | Area (km$^2$) | Sources |
|---|---|---|---|
| Gulf of Mexico | 180 | 4.4E + 5 | 59–61 |
| North Sea | 250 | 5.8E + 5 | 62,63 |
| Bengal | 66 | 3.2E + 4 | 64,65 |
| Yinggehai | 65 | 2.2E + 5 | 66–68 |
| Cuu Long | 25 | 8.4E + 4 | 69–71 |
| Nam Con Son | 28 | 2.5E + 5 | 70,71 |

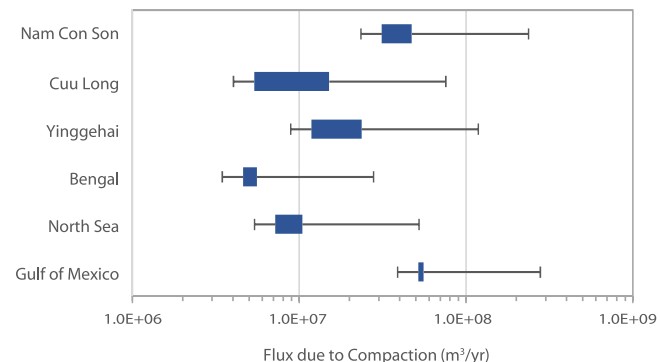

**Fig. 3 | Estimated fluxes from compacting basins.** Boxes span maximum and minimum values based on uncertainty in shale content. Whiskers indicate an additional factor of 2.1 uncertainty related to porosity and basin area. Source data are provided as a Source Data file.

## Table 3 | Basins likely to host geothermal convection

| Basin | Sediment type | Coastline length | Source(s) |
|---|---|---|---|
| Northwest Shelf Australia | Fine carbonates | 2500 km | 72,73 |
| Australian Bight | Med. carbonates | 1100 km | 74 |
| Celtic Sea | Med. carbonates | 840 km | 75 |
| North America: A[a] | Coarse clastics | 690 km | 76 |
| North America: B[b] | Coarse clastics | 830 km | 77 |
| North America: Florida | Med. carbonates | 580 km | 78 |
| Africa: A[c] | Fine carbonates | 2100 km | 79 |
| Africa: B[d] | Coarse clastics | 1300 km | 80 |
| Africa: C[e] | Fine carbonates | 890 km | 80 |
| South America: A[f] | Fine carbonates | 1700 km | 81,82 |
| South America: B[g] | Fine carbonates | 1600 km | 83 |

[a]Long Island – Virginia; [b]North Carolina – Georgia; [c]2°N–17°S; [d]16°N–26°N; [e]31°S–37°S; [f]33°S–46°S; [g]49°W–59°W.

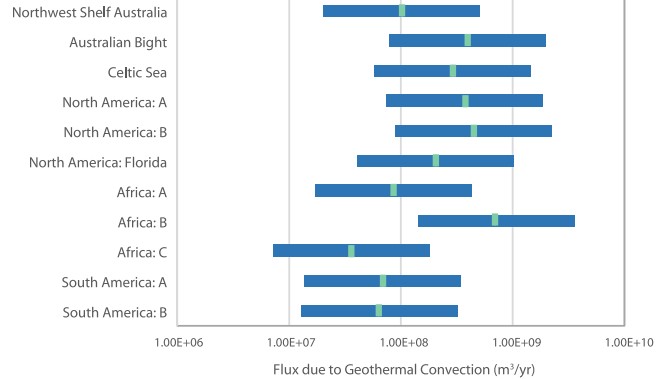

**Fig. 4 | Estimated geothermal fluxes from representative basins.** Green bars indicate best estimate based on simulations[19]. Blue boxes indicate a factor of five uncertainty above and below that estimate, based on uncertainty in the permeability of the basin. Source data are provided as a Source Data file.

to quickly assess (1) the compositional trends needed to balance the budget and (2) whether any of the archetypes could balance the budget in the absence of low-temperature basalt alteration.

The pattern of modifications in the theoretical fluid composition needed to balance marine chemical budgets in the absence of low-temperature basalt alteration showed sinks of K⁺, Mg²⁺, and Na⁺ and a source of Ca²⁺. This is consistent with the clastic-derived archetype, but

## Table 4 | Groundwater compositions of seawater, rivers, mid-ocean ridges (MORs), groundwater archetypes

| CONCENTRATIONS (mmol/L) | Ca²⁺ | K⁺ | Mg²⁺ | Na⁺ | Cl⁻ | Source(s) |
|---|---|---|---|---|---|---|
| Seawater | 10.3 | 10.3 | 53.4 | 47.0 | 547 | 6 |
| **Groundwater archetypes** | | | | | | |
| Cold seeps | 34.2 | 66.0 | 28.2 | 2640 | 2890 | 24,26,52 |
| Clastic-derived | 30.0 | 5.23 | 18.7 | 366 | 448 | 53,54 |
| Deep CaCl₂ brines | 1130 | 212 | 275 | 2100 | 5130 | 28ᵃ |
| Carbonate-derived | 34.3 | 23.3 | 101 | 1160 | 1310 | 56 |
| **Modified concentrations** | | | | | | |
| Modified cold seeps | −3.82 | 2.21 | −48.0 | 31.2 | 0 | - |
| Modified clastic-derived | 26.0 | −3.76 | −29.7 | −21.6 | 0 | - |
| Modified deep CaCl₂ brines | 110 | 12.3 | −23.9 | −246 | 0 | - |
| Modified carbonate-derived | 4.03 | −0.574 | −11.0 | 14.9 | 0 | - |
| Modified MORs | 28.9 | 28.1 | −52.7 | −39.5 | 0 | 6 |
| Modified rivers | 0.329 | 0.033 | 0.122 | 0.085 | 0 | 6 |

ᵃTable 8.

Modified concentrations have been normalized relative to modern seawater to indicate net contributions to seawater chemistry (see Section 4.4 and Supplementary Table 1 for methods).

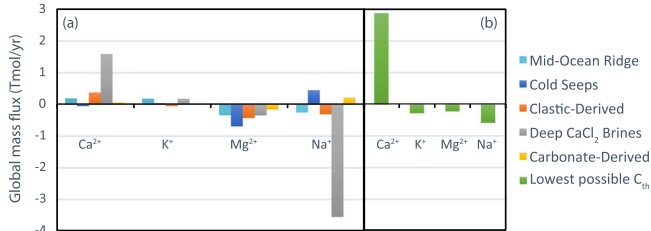

**Fig. 5 | Global ionic mass fluxes. a** Fluxes from mid-ocean ridge (MOR) hydrothermal convection and for the four fluid archetypes as trial fluxes from passive continental margins. **b** Minimum theoretical mass fluxes ($C_{th}$, Eq. 1) required to close chemical budgets if low-temperature basalt alteration is not considered. Source data are provided as a Source Data file.

this theoretical fluid showed unrealistic levels of alteration compared to seawater. Less alteration was required when we used our upper-bound estimate for the groundwater flux, but the degree of enrichment in Ca²⁺ still significantly exceeded that of any of the archetypes (Fig. 5b). These discrepancies strongly suggest that balancing marine chemical budgets requires contributions from low-temperature basalt alteration as well as groundwater discharge from passive margins.

### Global mass balance

We then calculated the average modified composition of low-temperature hydrothermal fluids ($C_{lt}$) that would be needed to balance the major ion budget of the ocean when our estimated passive margin fluxes are included, using an expanded mass balance equation:

$$Q_{mor}C_{mor} + Q_rC_r + Q_sC_s + Q_{lt}C_{lt} + M_C = 0 \qquad (2)$$

where $Q_{lt}$ is the volumetric flux of water through low-temperature hydrothermal systems, $0.6 \times 10^{13}$ to $2.0 \times 10^{13}$ m³ yr⁻¹ (ref. 11), and $C_s$ is the adjusted composition of groundwater discharging from passive margins. We created a range of compositions for $C_s$ using the clastic-derived and CaCl₂ brine archetypes, and we tested the range of volumetric fluxes for low-temperature hydrothermal fluids ($Q_{lt}$) suggested by Coogan and Gillis[11]. Using a range of compositions for

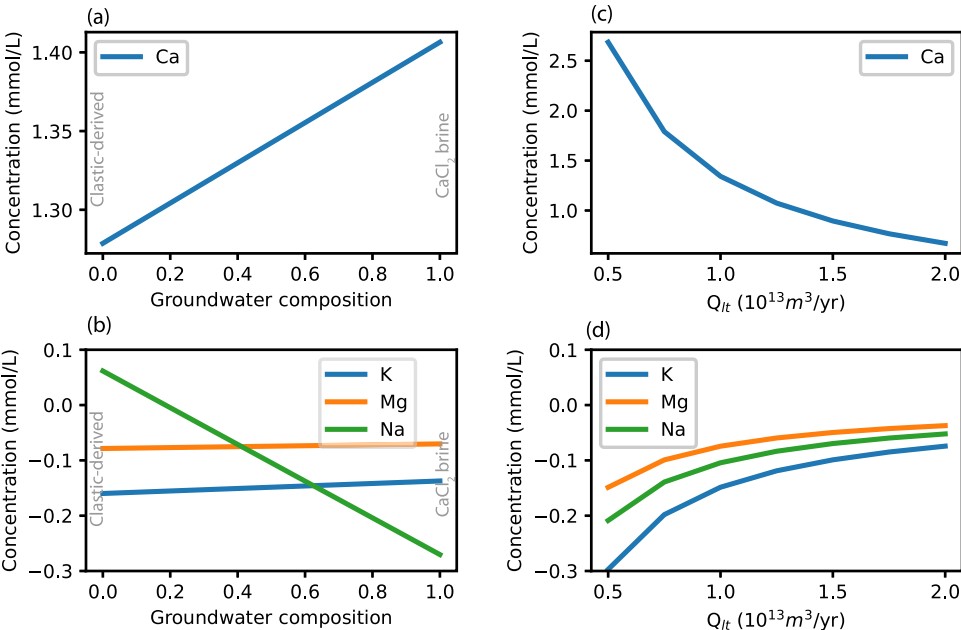

**Fig. 6 | Average adjusted composition of fluids from low-temperature basalt alteration ($C_{lt}$), calculated using Eq. 2. a** $Ca^{2+}$ component of $C_{lt}$ based on a range of compositions for groundwater discharging from passive margins ($C_s$). The x axis (Groundwater composition) indicates the fraction of $CaCl_2$ brines in two-component mixing between the clastic-derived and $CaCl_2$ brine archetypes. **b** $K^+$, $Mg^{2+}$, and $Na^+$ components of $C_{lt}$ for the conditions in **a**. **c** $Ca^{2+}$ component of $C_{lt}$ for $Q_{lt}$ ranging from 0.5 to 2.0 × 1013 $m^3$ $yr^{-1}$ (ref. 11). **d** $K^+$, $Mg^{2+}$, and $Na^+$ components of $C_{lt}$ for the conditions in **c**. Source data are provided as a Source Data file.

$C_s$ resulted in relatively small differences in $C_{lt}$ for $Ca^{2+}$, $Mg^{2+}$, and $K^+$, whereas the differences in $Na^+$ contributions between the clastic and $CaCl_2$ archetypes created larger variations in $Na^+$ (Fig. 6a, b). Our calculations showed that varying $Q_{lt}$ altered the degree of chemical alteration of $C_{lt}$ without changing the ratios of the ions with respect to one another (Fig. 6c, d). The resulting compositions for low-temperature basalt alteration are similar to observed fluid compositions at low-temperature basalt alteration sites (e.g., Dorado[39], 10–20 °C) for all elements except $Ca^{2+}$ (Supplementary Table 2), and the higher $Ca^{2+}$ concentrations can be achieved easily with a <3% contribution by fluids from higher temperature basalt alteration (e.g., Baby Bare[39], 60–65 °C).

## Variations in seawater chemistry over geologic time

The ratio of $Ca^{2+}$ to $Mg^{2+}$, $Na^+$, and $SO_4^{2-}$ in seawater has varied significantly over the Phanerozoic[3], with changes attributed mainly to variations in high-temperature MOR hydrothermal circulation and continental weathering. We suggest that changes in fluid and chemical fluxes through continental shelves contributed strongly to these variations. We specifically suggest a causal link between the significant low in continental shelf area 300–200 Ma[40]. and a major trough in the Ca/Mg ratio of seawater circa 250 Ma[41]. Consider also the large increase in the Ca/Mg ratio during the Cretaceous, when rising sea levels and temperatures drove an increase in carbonate platforms and shelves. Large fluxes through abundant, permeable carbonate platforms and associated dolomitization during this period would have sequestered significant $Mg^{2+}$ and while releasing equal moles of $Ca^{2+}$. This would have created a positive feedback loop, in which relatively high $Ca^{2+}$ concentrations supported increased carbonate deposition. Rapid growth of carbonate platforms in turn would support increased dolomitization and further release of $Ca^{2+}$. This feedback loop would only have been disrupted by falling sea levels, as the continents moved apart and the increasing average age of ocean crust allowed the depth of ocean basins to increase[3]. The rapid decrease in Ca/Mg ratios in seawater since the end of the Cretaceous is consistent with lower rates of geothermal convection during sea level lows[19] and a transition to lower-permeability clastic margins.

These findings also place the observed correlation between sea level and variations in seawater chemistry[3,42] in a new perspective. This link has so far appeared to be coincidental, in the sense that increases in MOR spreading lead to increases in the Ca/Mg ratio of seawater while also causing sea level to rise. We suggest a causal link instead, in which high sea levels increase fluid and chemical fluxes through passive margins, and these fluxes directly raise the Ca/Mg ratio in seawater.

## Other types of groundwater flow in continental shelves

Although we ignored submarine groundwater discharge (SGD) in our analysis, SGD may ultimately influence major ion chemistry and isotope systematics in the ocean. Studies have increasingly acknowledged the role of fresh SGD in Sr isotope ratios[29], but the volumetric flux of fresh SGD is <2% of river discharge globally[43,44]. In contrast, the volumetric flux of tide- and wind driven[45,46] saline (commonly 35 ppt) SGD rivals river discharge in the Atlantic basin[47]. We suggest that tide- and wind-driven saline SGD may be a candidate for future investigations of ocean chemistry. This type of SGD has much less time to react with sediments than geothermal convection or compaction-driven flow, but the sheer volume of flow in these systems has the potential to compensate for the relatively short residence time.

## Passive margins and marine isotope budgets

Continental shelves in passive margins act as huge flow-through reactors for saline water that affect the major ion chemistry of the ocean to a degree similar to that of high-temperature fluid circulation through MORs. Thus groundwater flow and reaction in continental shelves almost certainly affect marine isotope budgets as well. For example, recent mass balance calculations suggest that large volumes of dolomite must precipitate to balance Mg isotope budgets for the ocean[15]. Estimates of the contribution of fresh SGD to marine Mg isotope budgets[29] further increase the need for this sink. Although dolomite formation is extremely limited at low temperatures in shallow seafloor environments, dolomitization is a major sink for $Mg^{2+}$ in deep continental shelf settings. Thus, our model for flow and reaction in passive continental margin sediments solves the recent mystery of a

"hidden" dolomite sink for Mg[15] and provides a mechanism to resolve decades of speculation about additional sources and sinks for other ions[2,18,19]. Particularly as a new generation of isotope studies are emerging, it is critical that studies of the major ion chemistry and isotope budgets of the ocean consider chemical fluxes from continental shelves as well as rivers, mid-ocean ridges, and low-temperature basalt alteration.

## Methods

In this work, we estimated groundwater fluxes based on large-scale flow of groundwater with salinities similar to or exceeding seawater. Although fresh groundwater discharge has the potential to influence marine isotope budgets[29], we neglected fresh groundwater in the context of major ion budgets because its composition is similar to river water, and direct discharge of fresh groundwater to the ocean is less than 2% of global river discharge[44,48]. In contrast, fluids circulating through passive continental margins are significantly altered compared to the original seawater composition owing to sedimentary diagenetic reactions that occur over millions of years[27,28,49].

### Areas of Study

We considered 17 sedimentary basins in passive continental margins for this study (Fig. 2). We chose six basins with literature-verified overpressures (Table 2) to estimate compaction-driven flow. The overpressured basins were almost entirely composed of clastics (Supplementary Fig. 1), some with shale constituents greater than 90% (Supplementary Fig. 2). We assumed that basins that were dominated by coarse clastics and carbonates, and whose seafloor geometries resembled those in previous investigations of geothermal convection[19], were subject to geothermal convection (Table 3).

There are 105,000 km of passive margins on earth[30]. Based on estimates by Mouchet et al.[50] and our own measurements using Google Earth, there are $7.0 \times 10^6$ km² of overpressured submarine sediments along passive margins, which account for 24,000 km of coastline worldwide. We assumed that the remaining 81,000 km of passive margins had the potential to host geothermal convection. The six basins chosen to estimate compaction-driven flow in this study account for 23% of the earth's overpressured passive margin area. The 10 margins chosen to estimate geothermal convection in this study represent 17% of the 81,000 km. Given that both sets encompass roughly one fifth of the respective total lengths, the calculated groundwater fluxes were multiplied by a factor of 5 to provide an estimate for global groundwater flux.

### Groundwater discharge from compacting basins

Total pore-volume loss from the compacting basins was estimated by decompacting the layers in the stratigraphic columns (Supplementary Figs. 1 and 2) of the six continental shelf complexes in Table 2. The current porosity of each layer at depth was estimated based on generalized compaction curves from Baldwin and Butler[51] for sand and shale. For sand/sandstone,

$$\phi = 0.49 / e^{\left(\frac{d}{3.7}\right)} \qquad (3)$$

where $\phi$ is the porosity, $d$ is the depth in km, and e is Euler's number. For normally compacted mud/shale,

$$\phi = 1 - \sqrt[8]{d/15} \qquad (4)$$

For overpressured shale,

$$\phi = 1 - \sqrt[6.35]{d/6.02} \qquad (5)$$

To obtain a conservative estimate of compaction-driven flow, all sediments at depths below 200 m were assumed to be overpressured.

Assuming all sediments were normally compacted throughout the basins increased compaction by ~10%.

The sediments were decompacted to obtain the original thickness according to:

$$b_o = X_{sh} b \left( \frac{1 - \phi_{sh}}{1 - \phi_{sh}^o} \right) + X_{sand} b \left( \frac{1 - \phi_{sand}}{1 - \phi_{sand}^o} \right) \qquad (6)$$

where $b_o$ is the original thickness of the layer, $b$ is the current thickness of the layer, $X_{sh}$ is the percent shale composition of the layer, $X_{sand}$ is the percent sand composition of the layer, $\phi_{sh}$ is the porosity of shale at the depth of the layer, $\phi_{sh}^o$ is the original porosity of shale at deposition (60%), $\phi_{sand}$ is the porosity of sand at the depth of the layer, and $\phi_{sand}^o$ is the original porosity of sand (35%). This process was repeated for every layer in the stratigraphic column and summed to obtain the total original thickness. Fracturing in the sediment column was not mathematically considered because it does not affect the volume of the sediment. Cases of basin inversion and unconformities in the sedimentary record were also not considered in the calculations. Neglecting basin inversion and unconformities could lead to underestimates of the total fluxes from the basin.

The change in thickness from decompaction was then multiplied by the basin's area to calculate the total volume of fluid lost during compaction. The offshore area of each basin was estimated based on measurements from previous studies (Table 2) and confirmed by estimates using Google Earth (Supplementary Fig. 4). The resulting volume of compaction was divided by the age of the basin, yielding an average annual flux of groundwater ($Q_t$) since initial deposition:

$$Q_t = \frac{(b_o - b) * A}{dt} \qquad (7)$$

where $A$ is the basin's area and $dt$ is the basin's age. Calculating the discharge this way results in an average flux over time, whereas the rate of compaction in real sedimentary basins is unlikely to remain constant. For this reason we included basins of a variety of ages, and we consider these "average annual" fluxes to be very long-term (10 s of My) averages.

The estimated fluxes are subject to uncertainty from several sources, including uncertainty in basin area, porosity at depth, and sediment composition. We considered uncertainty in shale fractions by calculating fluid discharges for each continental shelf complex using upper and lower bound values for the shale fraction in each layer (Supplementary Fig. 2). The uncertainty in the initial porosity is unlikely to be more than a factor of two, and the estimated basin areas are probably accurate to within 10%. We applied these uncertainties to the upper and lower bounds from the shale fraction uncertainty to generate estimated maximum and minimum fluxes. As will be shown, the annual flux from geothermal convection proved to be much larger, reducing the impact of uncertainty in compaction-driven fluxes on our results.

### Groundwater discharge from geothermal convection

Geothermal fluxes were estimated based on *Wilson*[19], who used numerical models to estimate a range of possible groundwater fluxes due to geothermal convection using the bathymetry of continental shelves from the east coast of North America. Wilson used realistic seafloor geometries for seven cross-sections and reported a range of fluxes representative of five composite rock types. We calculated an average flux for each composite rock type by taking a geometric mean of the fluxes from the cross-sections. The dominant lithology of each shelf complex considered in this study was estimated from published cross-sections (Table 3), and the averaged representative flux from Wilson[19] was assigned to each basin and multiplied by the length of the coastline. The length of the coastline was estimated from the

publications in Table 3 and confirmed using Google Earth. We estimated that the uncertainty associated with applying a single lithology to an entire shelf complex was plus or minus a factor of five. This gives minimum and maximum flux values that differ by a factor of 25 for each basin.

## Groundwater archetypes and net chemical exchange

As previously indicated, the groundwater archetypes represent example fluid compositions across the spectrum of compositions observed in sedimentary basin settings. The first fluid archetype represents cold seeps, a global phenomenon of seepage of volatile elements and fluids from the seafloor. We chose example fluids based on observed discharge to the ocean in the form of seafloor brine pools in the Gulf of Mexico (GoM)[24,26,52]. They are enriched in $K^+$ and $Na^+$ and depleted in $Ca^{2+}$ and $Mg^{2+}$, and they have a high chlorinity (103,000 mg/l) relative to seawater. These fluids differ from cold seeps observed in convergent margins, which are typically depleted in $K^+$ via such reactions as the smectite-illite transition[33], and in gas hydrate settings, which also show depletion in $K^+$ refs. [37,38]. Discharge of such high-salinity fluids can only be driven by overpressures at depth, which are well-documented in the Gulf of Mexico. Depletion of $Ca^{2+}$ relative to seawater is a common feature of cold seep fluids in all of these settings and likely reflects precipitation of calcite during $CO_2$ degassing as the fluids emerged from high pressures at depth.

The second archetype represents fluids in basins dominated by clastic sediments. We compiled eight groundwater samples from depths of 350–400 m below the seafloor on the New Jersey continental shelf[53], an interval where high salinities (averaging ~75% of seawater salinity) indicate mild dilution of seawater by fresh groundwater during sea level low stands. We also compiled 17 samples from depths of 1500–4100 m below the land surface from the San Joaquin Basin[54], a well-studied basin that lacks carbonate sediment layers[36]. Salinities from San Joaquin samples averaged roughly 80% of seawater salinity, reflecting dilution by release of water during illitization[55]. Despite very different geological histories, these fluids show remarkably similar chemical trends. They are enriched in $Ca^{2+}$ and depleted in $K^+$, $Mg^{2+}$, and $Na^+$ relative to seawater.

The third fluid archetype was modeled after deep basinal $CaCl_2$ brine fluids[28], also known as oil-field brines. These fluids are representative of high salinity (five to 10 times the salinity of seawater), high density fluids with long residence times at elevated temperatures (>100 °C). The composition of the archetype is an arithmetic average of eight samples from the GoM, the Illinois Basin, the Michigan Basin, and the Dead Sea presented by Hardie[28] as representative $CaCl_2$ brines. As is common for this kind of deep basin brine, these fluids are enriched in $Ca^{2+}$ and $K^+$ and depleted in $Mg^{2+}$ and $Na^+$.

The final fluid archetype represents fluids in carbonate-dominated basins. This archetype was created by averaging ion concentrations of 26 samples (depths 2 to 409 mbsf) from the Great Australian Bight[56], which represents a spectrum from barely altered seawater at shallow depths to concentrated (three times seawater salinity) brines at depth, suggesting evaporation or interaction with evaporite sequences. These brines are enriched in $Ca^{2+}$ and $Na^+$ and depleted in $Mg^{2+}$ and $K^+$, clearly reflecting dolomitization. These trends are also consistent with trends from warm spring samples from the Floridan platform[57] for $Ca^{2+}$, $Mg^{2+}$, and $K^+$. The Floridan samples are slightly depleted in $Na^+$ and have a salinity roughly equal to seawater. We did not include the Floridan samples in the archetype calculations because they are significantly diluted by mixing with modern seawater.

We calculated net chemical fluxes for each input fluid based on the assumption that the net chloride flux is zero[6]. We first adjusted the compositions of all input fluids to zero chloride by subtracting ions in proportion to the composition of modern seawater. This approach assumes that seawater-derived salt aerosols dissolved in rainwater are the source of chloride in river waters[6] and that seawater was the

original source of water for seafloor hydrothermal convection and for our groundwater archetypes. Details of this process are shown in Supplementary Table 1.

Net chemical fluxes were calculated by multiplying the adjusted concentration of each archetype by the volumetric fluxes estimated above to generate a mass flux (Tmol $yr^{-1}$) for each ion. The resulting mass fluxes represent individual points in a range of possible groundwater compositions, as they each assume total discharge is representative of only one archetype.

## Mass balance model for the major ion chemistry of the ocean

The mass balances are represented in Eqs. (1) and (2) required fluxes from several sources other than our calculations. River chemical mass fluxes were gathered from Spencer and Hardie[6]. Chemical mass fluxes from high-temperature MOR hydrothermal flow were synthesized using the MOR brine composition chosen by Spencer and Hardie[6], from Reykjanes, Iceland, and the MOR axial fluid flux presented by Nielsen et al.[35], given the agreement between their upper bound estimates and the heat flow estimates of Mottl and Wheat[12]. Removal of ions due to precipitation of carbonates assumed a formula for high-Mg calcite of $(Ca_{0.9}, Mg_{0.1})CO_3$[58]. A recent review of estimated carbonate accumulation[10] suggests differences of no more than 20% from Milliman[2], so we used Milliman's estimate for the total carbonate accumulation rate, 3.2 billion tons per year. Fluid fluxes in low-temperature hydrothermal systems have been estimated based on energy balances[11], but the average composition of those fluids is highly uncertain. We first tested our model to ensure that it could replicate the results of *Spencer and Hardie*[6] before we updated it to use the volumetric hydrothermal fluid flux through MORs of Nielsen et al.[35] and added terms for passive margin and low-temperature hydrothermal fluxes. We note that the estimates of low-temperature hydrothermal volumetric fluxes include only off-axis flow, which excludes low-temperature axial flows[11].

## Data availability

The authors declare that the data supporting the findings of this study are available within the paper and its supplementary information files. The compaction, flux, and geochemical composition data generated in this study are provided in the Source Data file. Source data are provided with this paper.

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

## Acknowledgements
This material is based upon work supported by the National Science Foundation under Grant Nos. 1316250 (A.M.W. and S.M.W.) and 1736557 (A.M.W.).

## Author contributions
All three authors shared in the conceptualization of the work. A.O. did the original calculations, made the initial figures, and wrote the initial manuscript. A.M.W. provided funding and rewrote and expanded the scope of the work in the manuscript for the current journal. S.M.W. aided with manuscript and figure preparation, particularly with regard to M.O.R. and low-temperature basalt alteration.

## Competing interests
The authors declare no competing interests.
