## [Peer Review File · Nature Communications]

Large-scale groundwater flow and sedimentary diagenesis in continental shelves influence marine chemical budgetsREVIEWER COMMENTS

Reviewer #1 (Remarks to the Author):

The paper by Wilson et al. Estimated the flux of ions to the ocean through groundwater circulation from continental shelves of passive margins that is driven by both sediment compaction and geothermal convection. The authors claim that, such a flux, together with low-temperature basalt alteration, has the potential to close the budget for several major cations in the ocean. While it is exciting to see studies like this put diagenetic fluxes into the discussion of marine cation budget, I found the study not convincing enough given the loose constraints for the fluid composition. The authors have gone through a comprehensive effort to derive volumetric fluxes for 17 different basin and show that the overall flux is at least compatible to that from high temperature MOR convection and could very likely be higher. I think this finding is significant and well constrained. However, the authors didn't seem to dedicate the same amount of effort to constrain fluid composition. For example, why four archetype? Why the fluid composition from those few locations? Are they representative enough for a global scale estimation? I happen to have done quite a bit of work from the cold seeps along passive margins (northern Norwegian margin, Barents Sea, and Fram Strait. See later for selected papers. It is not my intention to press for citation but simply showing what has been done) and we have documented very different fluid composition from the that of GoM. For example, we see that these cold seep fluids contain less potassium than seawater (ie K sink). In addition, these cold seeps are not such a strong Mg sink as the authors claimed (ie still a Mg sink but of very different magnitude). Arguably GoM is a very unique fluid system when comparing to any cold seeps around the world and it needs further justification why this site was thought to be representative by the authors. I don't have enough experiences to critique the authors' choices on the end-member of fluid composition for the other archetypes. Nonetheless, I notice the choice is only based on a handful of locations (2 locations for type 2, 4 locations from type 3 and 1 location for type 4). This makes me wonder how representative are these end-member compositions overall. I wonder if the authors can compile pore fluid data from ocean drilling programs (eg ODP and IODP) which should have a much better coverage of fluid composition data. Similar kind of applications can be found from the list included at the end.

I have another minor comment. It will be very nice if the authors can be more specific about the kind of "groundwater" they are talking about in the introduction. To me, the authors were dealing with circulation of seawater that is further modified by diagenetic processes. The term "groundwater" they used is a bit confusing to me, though I understand that, mechanistically speaking, the fluid focus here falls into the category of saline groundwater. However, later in the discussion, the authors also tried to distinguish the fluid they discussed in the paper with the saline SGD that transports Ra, which makes the actual definition of "groundwater" in this paper more confusing.

Dr. Wei-Li Hong
Stockholm University

Relevant papers for cold seep fluid composition from passive margins:

Hong, W.-L. et al. Seepage from an arctic shallow marine gas hydrate reservoir is insensitive to momentary ocean warming. *Nat. Commun.* 8, 15745 doi: 10.1038/ncomms15745 (2017).

Hong, W.-L., Lepland, A., Himmler, T., Kim, J.-H., Chand, S., Sahy, D., et al. (2019). Discharge of meteoric water in the eastern Norwegian Sea Since the last glacial period. *Geophysical Research Letters*, 46, 8194–8204. <https://doi.org/10.1029/2019GL084237>

Hong, W.-L., Thomas Pape, Christopher Schmidt, Haoyi Yao, K. Wallmann, Andreia Plaza-Faverola, J. W. B. Rae, Aivo Lepland, Stefan Bünz, and Gerhard Bohrmann. "Interactions between deep formation fluid and gas hydrate dynamics inferred from pore fluid geochemistry at active pockmarks of the Vestnesa Ridge, west Svalbard margin." *Marine and Petroleum Geology* 127 (2021): 104957.

Examples for the use IODP data:

Sun, Xiaole, and Alexandra V. Turchyn. "Significant contribution of authigenic carbonate to marine carbon burial." *Nature Geoscience* 7, no. 3 (2014): 201-204.

Berg, Richard D., Evan A. Solomon, and Fang-Zhen Teng. "The role of marine sediment diagenesis in the modern oceanic magnesium cycle." *Nature communications* 10, no. 1 (2019): 4371.

Reviewer #2 (Remarks to the Author):

In their manuscript, Wilson et al. propose and demonstrate that offshore submarine groundwater discharge from continental shelf sediments in passive margins can account for missing sources and sinks in global marine chemical budgets for calcium (Ca), magnesium (Mg) and potassium (K). They integrate a combination of calculations, observations, and models from past studies, focusing their efforts around quantifying chemical sources/sinks for two hydrologic processes: compaction-driven dewatering in areas of rapid sedimentation and geothermal convection. This paper could be an important contribution to marine geochemistry and geophysics communities. It is novel in that it offers a fresh take on longstanding geochemical budgets. It also may open the door to interpreting new processes from old paleochemistry measurements in seafloor sediments. The analysis appears technically sound, but the methods are rather thin or vague in places, and the results are difficult to digest in tabular form. I offer the following suggestions for clarifying methods and results.

Equation 4 is a key equation, and the tables and figures in the main text all lead towards populating unknowns in the equation, but it is hard to put together the pieces until the reader finishes the methods. I'd suggest presenting the equation sooner with a graphical illustration of the terms. This can be done by expanding Figure 1 to show rivers and MOR's (the full cycle and fluxes for the elements of interest). A

more detailed conceptual figure with all the processes that go into the geochemical budgets in Equation 4 would also help clarify some of the ideas in L181-194. This figure and the overall paper would also be clearer for non-geochemists if reaction equations were shown for dolomitization, illitization, and albitization. For example, this would clarify the statement in L 66.

The calculation of fluxes for compaction-driven flow left much to be accepted on faith. I see that this process contributes less to Q_s than geothermal convection and therefore would not change the results much, even if the numbers changed. That said, the inputs to Table 1 should be described in more detail in a supplement. For example, all of the Gulf of Mexico is not undergoing similar sedimentation rates and is not subject to the same degree of compaction-driven dewatering. How does the area estimate, which appears to be approximated as a rectangle, account for this? It would help to include a supplement with a couple of paragraphs and a basin-scale map for each basin, along with a more detailed explanation of the datasets used in the parameter estimates. One example of a supplement that has more detailed paragraphs on different regions of analysis can be found in Befus et al. (2017). The same suggestion could also be made for the basins subject to geothermal fluxes. For example, “length of coastline” in L 337 may not be the same as basin length, especially if coastline length includes all its inlets and peninsulas. These are relatively small uncertainties either way that should not substantially modify the results, but it is still important to understand what the authors have measured and reported.

I am still fuzzy on the porosity or lithology sequences that were decompacted. The authors cite Baldwin and Butler’s compaction curves—are they just using the idealized curves? What stratigraphic columns did they use (L 272, LL 299)? The authors mention shale fractions for each layer, but I’m not sure what layers or datasets they refer to without a stratigraphic column or similar figure. This is an example where it would help to have more graphical information in a supplement.

Compaction-driven flow also exists in tectonic basins, but this is not shown to be small until L 248. Consider stating sooner.

Equation 4 and the geochemical analyses are difficult to envision, especially using tables. This information would be better presented graphically, perhaps through ternary diagrams showing the 4 end-members. This could also help graphically explain statements like L 127. The authors would also do well to present (using smaller symbols or at least boxes) the scatter in samples from the cited papers for each water type (Table 3) to show how well the chosen archetypes represent the natural variability in each water type.

Figures 3 and 4: I would also be keen to see this information in m/y , to account for differences in the map view area of each basin. In general, this would help clarify the relative punch that compacting basins provide—maybe they are small but mighty for their size when it comes to the total fluid they expel?

L 57: By volume or map area? The latter seems more relevant to this statement.

L 95: I think the authors are suggesting that the flux in the GoM is higher because shales can compact

more, but as mentioned earlier, it would also depend on deposition rates and ages. Please add something like "...shale content and deposition rate."

Table 5 is difficult to follow, as units for most columns are in the caption, but units for the left column are in the column header. Also, the term off-axis isn't really introduced until Table 5, adding some uncertainty in terminology.

L 216: This conclusion can be driven home with an improved conceptual figure (Figure 1).

L 242: Rather than citing a regional study from Befus et al. for a global estimate, consider citing Zhou et al.

Befus, K. M., Kroeger, K. D., Smith, C. G., & Swarzenski, P. W. (2017). The magnitude and origin of groundwater discharge to eastern U.S. and Gulf of Mexico coastal waters. *Geophysical Research Letters*, 44, 10,396–10,406. <https://doi.org/10.1002/2017GL075238>

Zhou, Y. Q., Sawyer, A. H., David, C. H., & Famiglietti, J. S. (2019). Fresh submarine groundwater discharge to the near-global coast. *Geophysical Research Letters*, 46. <https://doi.org/10.1029/2019GL082749>

Response to Reviewer's Comments

These reviewer comments were very helpful, particularly in realizing that the audience for offshore groundwater flow has evolved significantly over the years to now include many researchers who are not familiar with regional-scale groundwater flow in sedimentary basins. We made a serious effort to clarify our terminology for an audience with a broader range of perspectives. We also included the Source Data file with this submission, which was accidentally omitted last time. At the excellent suggestion of Reviewer 2 we also completely redrafted Figure 1, converted tables 4&5 to a single figure (now Figs 5a and b), and made quite a number of other new supplemental figures.

In addition to changes suggested by the reviewers, we edited the paper to fit Nature Communications guidelines. This primarily involved moving material from the last paragraph of the introduction to the methods, to avoid having figures other than the conceptual model in the Introduction.

Line numbers refer to the clean copy rather than the track-changes version, which was messy.

Reviewer #1 (Remarks to the Author):

1. The paper by Wilson et al. Estimated the flux of ions to the ocean through groundwater circulation from continental shelves of passive margins that is driven by both sediment compaction and geothermal convection. The authors claim that, such a flux, together with low-temperature basalt alteration, has the potential to close the budget for several major cations in the ocean. While it is exciting to see studies like this put diagenetic fluxes into the discussion of marine cation budget, I found the study not convincing enough given the loose constraints for the fluid composition. The authors have gone through a comprehensive effort to derive volumetric fluxes for 17 different basins and show that the overall flux is at least compatible to that from high temperature MOR convection and could very likely be higher. I think this finding is significant and well constrained. However, the authors didn't seem to dedicate the same amount of effort to constrain fluid composition. For example, why four archetypes? Why the fluid composition from those few locations? Are they representative enough for a global scale estimation?

> We should have done a better job explaining the choice of four archetypes in the original draft. We have now clarified this starting line 148, and we added Supplementary Figure 3 to the supplement to show how the archetypes compare with other groundwaters in sedimentary basins. New wording at line 87 is:

"We considered four example fluid compositions to estimate possible chemical mass fluxes, as a proof of concept and to inform later mass balance calculations. The four examples, hereafter referred to as archetypes, are not endmembers but rather representative of the continuum of groundwater/brine compositions³⁰ encountered in deep sedimentary basin settings³⁰ (Supplementary Figure 3), including representative compositions from (1) observed saline cold seeps, (2) clastic-dominated basins, (3) deep CaCl₂ brines²³, and (4) carbonate-dominated basins (Table 4). The cold seep archetype comes from observed brine seeps in the Gulf of Mexico. Deep CaCl₂ brines represent highly evolved fluids characterized by high-temperatures (>100°C), high salinities, and long reaction times (millions of years). The clastic- and carbonate-dominated archetypes represent specific sediment compositions, in which dolomitization is excluded (clastic) or is the dominant form of diagenesis (carbonate).

2. I happen to have done quite a bit of work from the cold seeps along passive margins (northern Norwegian margin, Barents Sea, and Fram Strait). See later for selected papers. It is not my intention

to press for citation but simply showing what has been done) and we have documented very different fluid composition from the that of GoM. For example, we see that these cold seep fluids contain less potassium than seawater (ie K sink). In addition, these cold seeps are not such a strong Mg sink as the authors claimed (ie still a Mg sink but of very different magnitude). Arguably GoM is a very unique fluid system when comparing to any cold seeps around the world and it needs further justification why this site was thought to be representative by the authors. I don't have enough experiences to critique the authors' choices on the end-member of fluid composition for the other archetypes.

> Dr. Hong is correct that our cold seeps are very different from (and more highly reacted than) the cold seeps he has documented in other settings. In addition to clarifying our intentions for the archetypes (see comment #1), we also added this clarification in the 4th paragraph (line 46), to let readers know much earlier that we are considering saline groundwater rather than fresh groundwater:

Note that the groundwater that participates in sedimentary diagenesis in continental shelves is overwhelmingly saline (of seawater salinity or higher, including brines), originating as seawater rather than rainwater.

I appreciate learning of Dr. Hong's papers about fresh and brackish seeps. I know of another researcher who will be very interested to have these references, and I will certainly share them with my students.

3. Nonetheless, I notice the choice is only based on a handful of locations (2 locations for type 2, 4 locations from type 3 and 1 location for type 4). This makes me wonder how representative are these end-member compositions overall. I wonder if the authors can compile pore fluid data from ocean drilling programs (eg ODP and IODP) which should have a much better coverage of fluid composition data. Similar kind of applications can be found from the list included at the end.

This is now addressed in the revised discussion of the groundwater archetypes and Supplementary Figure 3 (see #1). The new figures make it clear, for example, that the single citation for type 4 actually represents a compilation from 8 well-studied sites. The new figures also show the archetypes in the context of broader compilations.

Regarding IODP datasets: We used IODP data for our clastic and carbonate archetypes, but these data are rare. Continental shelves are not well sampled by IODP because most of the relevant areas are in waters too shallow for their dedicated ships. (The Resolution was not equipped to handle overpressures and other dangerous drilling conditions in the Gulf of Mexico, either.) (This is beside the point of the paper, so we do not go into this in the text, but this gap in IODP data makes it very difficult to know how to interpret Sun and Turchyn 2014 or Berg et al. 2019.)

4. I have another minor comment. It will be very nice if the authors can be more specific about the kind of "groundwater" they are talking about in the introduction. To me, the authors were dealing with circulation of seawater that is further modified by diagenetic processes. The term "groundwater" they used is a bit confusing to me, though I understand that, mechanistically speaking, the fluid focus here falls into the category of saline groundwater. However, later in the discussion, the authors also tried to distinguish the fluid they discussed in the paper with the saline SGD that transports Ra, which makes the actual definition of "groundwater" in this paper more confusing.

As indicated in comments #1 and #2, we revised the wording early on to clarify the kind of groundwater we are interested in in this paper. As for the discussion, we simplified this paragraph by simply

referring to the driving force for flow (tide- and wind-driven vs. compaction-driven or geothermal flow). It makes the point without getting tangled up in what is or is not “SGD”:

“In contrast, the volumetric flux of tide- and wind-driven^{39,40} saline (commonly 35 ppt) SGD rivals river discharge in the Atlantic basin⁴¹. We suggest that tide- and wind-driven saline SGD may be a candidate for future investigations of ocean chemistry. This type of SGD has much less time to react with sediments than geothermal convection or compaction-driven flow, but the sheer volume of flow in these systems has the potential to compensate for the relatively short residence time.”

Dr. Wei-Li Hong
Stockholm University

Relevant papers for cold seep fluid composition from passive margins:

Hong, W.-L. et al. Seepage from an arctic shallow marine gas hydrate reservoir is insensitive to momentary ocean warming. *Nat. Commun.* 8, 15745 doi: 10.1038/ncomms15745 (2017).

Hong, W.-L., Lepland, A., Himmler, T., Kim, J.-H., Chand, S., Sahy, D., et al. (2019). Discharge of meteoric water in the eastern Norwegian Sea Since the last glacial period. *Geophysical Research Letters*, 46, 8194–8204. <https://doi.org/10.1029/2019GL084237>

Hong, W.-L., Thomas Pape, Christopher Schmidt, Haoyi Yao, K. Wallmann, Andreia Plaza-Faverola, J. W. B. Rae, Aivo Lepland, Stefan Bünz, and Gerhard Bohrmann. "Interactions between deep formation fluid and gas hydrate dynamics inferred from pore fluid geochemistry at active pockmarks of the Vestnesa Ridge, west Svalbard margin." *Marine and Petroleum Geology* 127 (2021): 104957.

Examples for the use IODP data:

Sun, Xiaole, and Alexandra V. Turchyn. "Significant contribution of authigenic carbonate to marine carbon burial." *Nature Geoscience* 7, no. 3 (2014): 201-204.

Berg, Richard D., Evan A. Solomon, and Fang-Zhen Teng. "The role of marine sediment diagenesis in the modern oceanic magnesium cycle." *Nature communications* 10, no. 1 (2019): 4371.

Reviewer #2 (Remarks to the Author):

1. In their manuscript, Wilson et al. propose and demonstrate that offshore submarine groundwater discharge from continental shelf sediments in passive margins can account for missing sources and sinks in global marine chemical budgets for calcium (Ca), magnesium (Mg) and potassium (K). They integrate a combination of calculations, observations, and models from past studies, focusing their efforts around quantifying chemical sources/sinks for two hydrologic processes: compaction-driven dewatering in areas of rapid sedimentation and geothermal convection. This paper could be an important contribution to

marine geochemistry and geophysics communities. It is novel in that it offers a fresh take on longstanding geochemical budgets. It also may open the door to interpreting new processes from old paleochemistry measurements in seafloor sediments. The analysis appears technically sound, but the methods are rather thin or vague in places, and the results are difficult to digest in tabular form. I offer the following suggestions for clarifying methods and results.

> Let me apologize here: I accidentally left the “Source Data” spreadsheet out of the original submission. It is now included. Meanwhile, the suggestion to turn more of the tables into figures is an excellent one. We definitely followed this suggestion, as detailed below.

2. Equation 4 is a key equation, and the tables and figures in the main text all lead towards populating unknowns in the equation, but it is hard to put together the pieces until the reader finishes the methods. I’d suggest presenting the equation sooner with a graphical illustration of the terms. This can be done by expanding Figure 1 to show rivers and MOR’s (the full cycle and fluxes for the elements of interest). A more detailed conceptual figure with all the processes that go into the geochemical budgets in Equation 4 would also help clarify some of the ideas in L181-194. This figure and the overall paper would also be clearer for non-geochemists if reaction equations were shown for dolomitization, illitization, and albitization. For example, this would clarify the statement in L 66.

> This was a great suggestion. We redrafted Figure 1. None of the prominent papers on marine major ion chemistry that I have seen show conceptual diagrams (including all the recent ones published in this journal), but now that we have one I like it quite a lot. We put Table 1 (stoichiometric equations for sedimentary diagenetic reactions) back in the paper. We moved the mass balance equations up (now lines 179 and 209). Thanks to the reviewer here.

3. The calculation of fluxes for compaction-driven flow left much to be accepted on faith. I see that this process contributes less to Q_s than geothermal convection and therefore would not change the results much, even if the numbers changed. That said, the inputs to Table 1 should be described in more detail in a supplement. For example, all of the Gulf of Mexico is not undergoing similar sedimentation rates and is not subject to the same degree of compaction-driven dewatering. How does the area estimate, which appears to be approximated as a rectangle, account for this? It would help to include a supplement with a couple of paragraphs and a basin-scale map for each basin, along with a more detailed explanation of the datasets used in the parameter estimates. One example of a supplement that has more detailed paragraphs on different regions of analysis can be found in Befus et al. (2017). The same suggestion could also be made for the basins subject to geothermal fluxes. For example, “length of coastline” in L 337 may not be the same as basin length, especially if coastline length includes all its inlets and peninsulas. These are relatively small uncertainties either way that should not substantially modify the results, but it is still important to understand what the authors have measured and reported.

> We digitized these using Google Earth, as now explained in more detail (line 367 and Supplementary Figure 4). We resisted showing diagrams of each basin, given the simplicity of this operation and the fact that a quick look at any of the references in Table 2 gives a good idea of the extent of any given basin, but we showed an example from the Gulf of Mexico in the Supplement, to illustrate the degree of detail

(not very high, given that we were only going for two significant figures).

4. I am still fuzzy on the porosity or lithology sequences that were decompacted. The authors cite Baldwin and Butler's compaction curves—are they just using the idealized curves? What stratigraphic columns did they use (L 272, LL 299)? The authors mention shale fractions for each layer, but I'm not sure what layers or datasets they refer to without a stratigraphic column or similar figure. This is an example where it would help to have more graphical information in a supplement.

> The stratigraphic datasets are in the Source Data, and we plotted strat columns and shale content in the Supplement (new Supplementary Figures 1 and 2).

5. Compaction-driven flow also exists in tectonic basins, but this is not shown to be small until L 248. Consider stating sooner.

> We moved this statement (L248) from the methods to the introduction (line 64).

6. Equation 4 and the geochemical analyses are difficult to envision, especially using tables. This information would be better presented graphically, perhaps through ternary diagrams showing the 4 end-members. This could also help graphically explain statements like L 127. The authors would also do well to present (using smaller symbols or at least boxes) the scatter in samples from the cited papers for each water type (Table 3) to show how well the chosen archetypes represent the natural variability in each water type.

> Good suggestions. We moved Table 4 to the supplement and replaced it with a graph (Figure 5a) showing the archetype fluxes relative to MOR fluxes, to demonstrate the likely importance of the groundwater contribution. We also removed Table 5, replacing it with a graph showing the theoretical groundwater composition needed to close the budgets in the absence of low-temperature basalt alteration, to identify important trends (Figure 5b). This allowed us to rewrite the Chemical Mass Fluxes section to be much simpler (including the former L 127, which needed simplifying), and this section now incorporates what was previously the first paragraph or so of the Global Mass Balance section. The archetypes are now shown in the context of earlier (1994) compilations of basin fluids in Supplementary Figure 3.

7. Figures 3 and 4: I would also be keen to see this information in m/y, to account for differences in the map view area of each basin. In general, this would help clarify the relative punch that compacting basins provide—maybe they are small but mighty for their size when it comes to the total fluid they expel?

>This was a good question, which we now address on line 134.

“Although the fluxes per kilometer of coastline (or of basin length, for most overpressured basins) are of the same order of magnitude for geothermal convection and sediment compaction, compaction-driven flow comprises just 5% of the global flow. This is largely because compaction-driven flow is relatively uncommon in continental shelves”

L 57: By volume or map area? The latter seems more relevant to this statement.

> now clarified on line 62 “(by area and volume)”

L 95: I think the authors are suggesting that the flux in the GoM is higher because shales can compact more, but as mentioned earlier, it would also depend on deposition rates and ages. Please add something like “...shale content and deposition rate.”

> done

Table 5 is difficult to follow, as units for most columns are in the caption, but units for the left column are in the column header. Also, the term off-axis isn't really introduced until Table 5, adding some uncertainty in terminology.

> Table 5 has been replaced, see #6 above. (Off-axis was supposed to have been changed to low-temperature -- I think we eradicated the rest of them.)

L 216: This conclusion can be driven home with an improved conceptual figure (Figure 1).

> done (above)

L 242: Rather than citing a regional study from Befus et al. for a global estimate, consider citing Zhou et al.

> done, another great suggestion from this reviewer

Befus, K. M., Kroeger, K. D., Smith, C. G., & Swarzenski, P. W. (2017). The magnitude and origin of groundwater discharge to eastern U.S. and Gulf of Mexico coastal waters. *Geophysical Research Letters*, 44, 10,396–10,406. <https://doi.org/10.1002/2017GL075238>

Zhou, Y. Q., Sawyer, A. H., David, C. H., & Famiglietti, J. S. (2019). Fresh submarine groundwater discharge to the near-global coast. *Geophysical Research Letters*, 46. <https://doi.org/10.1029/2019GL082749>

REVIEWER COMMENTS

Reviewer #1 (Remarks to the Author):

In the previous round of review, I criticised mainly on the choices of representative fluid composition for the archetypes that is then used for their mass balance calculation. While I appreciate the additional efforts from Wilson et al. to accommodate my comments, such as including supplementary figure 3 to compare the composition of the four selected archetypes with other saline fluids in sedimentary basin and the additional paragraphs explaining archetypes, the replies are not entirely satisfactory to be honest.

Wilson et al. replied that “Note that the groundwater that participates in sedimentary diagenesis in continental shelves is overwhelmingly saline (of seawater salinity or higher, including brines), originating as seawater rather than rainwater.” They thus maintained their selection of using Gulf of Mexico as the case representing the “cold seep” archetype. Such a statement is confusing to me as, in their own selection of fluid composition for archetype “clastic-derived”, the salinity is apparent lower than seawater (448 mM vs. 547 mM of seawater), which apparently contradicts to the statement above. Fresher porewater doesn’t necessarily indicate mixing of meteoric water. Reactions such as clay dehydration, which is common in marine environment, can also result in fresher porewater as compared to seawater. In addition, the cold seep site I suggested for the northern Atlantic Ocean margin has a large seawater component with salinity only 10% lower than seawater value, which is comparable to the archetype “clastic-derived” selected by the authors. Again, I don’t think it is needed for the authors to include the site suggested. I just want to point out the large variation in fluid composition and the challenge to generalize.

Despite my agreement of the author’s point. I need to point out their selection of cold seep archetype composition will have minimum impact on the overall conclusion given that the authors only considered the most extreme composition for their global mass balance calculation. Cold seep composition is well within that range regardless of their choices of representative sites for cold seep archetype.

In my another comment for the fluid composition for the carbonate archetype, Wilson et al. replied the following: “This is now addressed in the revised discussion of the groundwater archetypes and Supplementary Figure 3 (see #1). The new figures make it clear, for example, that the single citation for type 4 actually represents a compilation from 8 well-studied sites. The new figures also show the archetypes in the context of broader compilations.” I did check back to the reference mentioned (ref 49 in the paper), which is the site report from ODP Leg182 Site 1126 (Great Australian Bight). I’m not sure why Wilson et al. indicated the data were from “8 well-studied site”. To me, this is data from several depths of a single location (on a side note, this drilling site has a water depth of 738.8 meters, which is not from continental shelf). Again, I’m not entirely convinced this could be representative for all the carbonate archetype for global sedimentary basins. The authors need to clarify this further. The composition of this archetype may be crucial to the calculation in Eqs. 1 and 2 and will deserve a really careful consideration.

I also have the following minor comments to help improve the readability of the paper.

Line 20-24: References should be included to support the statements in this paragraph

Line 33-35: this statement also needs to be backed up with references.

Line 36-37: This statement also needs support.

Line 99: GoM needs to be spelled out

Table 2: Why not have the full name for GoM?

Fig.4: To avoid confusion, I suggest change the title of x-axis to "flux due to geothermal convection", which is also consistent to the title of Fig. 3.

Line 122-124: Can this "predictability" be read from any figure or suppl. Material? Or is it really an important statement that needs to be included?

Table 4: Though the calculation for "modified concentration" is not included in the main text, a sentence should still be included to guide the readers to the calculation (such as section 4.4 of the method).

Supplementary table 1: Thanks for including detailed calculation here. Why items 4, 5, 9, 10 were indicated as flux, despite the unit is in concentration? Isn't that you simplify calculate the difference between seawater and the adjusted concentration from the brine?

Line 181, 183, 185: the references 31, 3,1 are easily confused with the exponent. I suggest write "(ref 3)" instead.

Line 184: the m from "mc" should be capitalized to be consistent with Eq .1. Refer C_{mor}, C_r, and mc (or Mc) back to table 4 and also include these acronyms in Table 4 to facilitate reading.

Line 185: is "Q_s" the same as Q_i in Line 132? If yes, use only one of them for consistency. If not, specify the value.

Line 197: "but it showed unrealistic level of seawater modifications" I don't understand this sentence. Which "it" was referred to and why seawater modification? I thought seawater has been corrected for the archetypes?

Line 198: Again, not quite sure what does seawater modification mean here.

Figure 5: indicated Eq. 1 in the caption to help the readers connect.

Eq. 2: is the "mc" in the equation the same as "Mc" in Eq. 1? Is the Q_s the same as Q_i in Line 132? Same comment as for Line 184. Refer back to Table 4 when calling the different terms. What is the value of C_{it} used? I cant see it in Table 4.

Line 214: reference 32 can be confused with exponent

Figure 6: Where is Eq. 4 indicated in the caption?

Line 218: I'm a bit confused here. So C_{it} was being calculated from C_s, which you adopted the composition of the clastic and CaCl₂ types? This could probably be already indicated in Line 211-214.

Reviewer #2 (Remarks to the Author):

I have reviewed the revised submission of Wilson et al. and find the revisions to be comprehensive and

effective. This is an exciting contribution that ties together a lot of lines of evidence to solve a fundamental question about marine geochemical budgets and hydrogeologic processes. A few very minor remaining suggestions:

- In the new conceptual figure (1A), it would help to label the fluxes (i.e. "QrCr next to 1, etc).
- Figures 3 and 4 could be combined (perhaps with the compaction-driven fluxes shown in one color above, and the geothermal fluxes in another color below).
- I could not find a reference to the source of information on lithology (Supplementary Figures 1 and 2). Was it the Baldwin and Butler reference? If so, I'd suggest putting this citation in the captions.

Response to Reviewers

REVIEWER COMMENTS

Reviewer #1 (Remarks to the Author):

In the previous round of review, I criticised mainly on the choices of representative fluid composition for the archetypes that is then used for their mass balance calculation. While I appreciate the additional efforts from Wilson et al. to accommodate my comments, such as including supplementary figure 3 to compare the composition of the four selected archetypes with other saline fluids in sedimentary basin and the additional paragraphs explaining archetypes, the replies are not entirely satisfactory to be honest.

Wilson et al. replied that “Note that the groundwater that participates in sedimentary diagenesis in continental shelves is overwhelmingly saline (of seawater salinity or higher, including brines), originating as seawater rather than rainwater.” They thus maintained their selection of using Gulf of Mexico as the case representing the “cold seep” archetype. Such a statement is confusing to me as, in their own selection of fluid composition for archetype “clastic-derived”, the salinity is apparent lower than seawater (448 mM vs. 547 mM of seawater), which apparently contradicts to the statement above. Fresher porewater doesn’t necessarily indicate mixing of meteoric water. Reactions such as clay dehydration, which is common in marine environment, can also result in fresher porewater as compared to seawater. In addition, the cold seep site I suggested for the northern Atlantic Ocean margin has a large seawater component with salinity only 10% lower than seawater value, which is comparable to the archetype “clastic-derived” selected by the authors. Again, I don’t think it is needed for the authors to include the site suggested. I just want to point out the large variation in fluid composition and the challenge to generalize.

Good point, thank you for being persistent. We got distracted by the meteoric water paper and did not realize that the other papers did indeed describe fluids of seawater origin that have reacted to some degree with sediments. We added a sentence (line 156-159): “These relatively highly-reacted fluids differ somewhat from cold seep fluids in other settings, including gas hydrate systems^{37,38} [refs are Hong et al. 2017; 2021] and convergent margins³⁰, but a key feature of all of these seeps is that they are depleted in Ca²⁺ relative to seawater.” On line 48 we changed “of seawater salinity” to “near seawater salinity.” We had already noted that San Joaquin fluids were diluted by clay dehydration reactions in the Methods (line 440).

Despite my agreement of the author’s point. I need to point out their selection of cold seep archetype composition will have minimum impact on the overall conclusion given that the authors only considered the most extreme composition for their global mass balance calculation. Cold seep composition is well within that range regardless of their choices of representative sites for cold seep archetype.

Another good point.

In my another comment for the fluid composition for the carbonate archetype, Wilson et al. replied the following: “This is now addressed in the revised discussion of the groundwater archetypes and Supplementary Figure 3 (see #1). The new figures make it clear, for example, that the single citation for

type 4 actually represents a compilation from 8 well-studied sites. The new figures also show the archetypes in the context of broader compilations.” I did check back to the reference mentioned (ref 49 in the paper), which is the site report from ODP Leg182 Site 1126 (Great Australian Bight). I’m not sure why Wilson et al. indicated the data were from “8 well-studied site”. To me, this is data from several depths of a single location (on a side note, this drilling site has a water depth of 738.8 meters, which is not from continental shelf). Again, I’m not entirely convinced this could be representative for all the carbonate archetype for global sedimentary basins. The authors need to clarify this further. The composition of this archetype may be crucial to the calculation in Eqs. 1 and 2 and will deserve a really careful consideration.

Our apologies, there was a typo in the letter last time. The single citation with 8 well-studied sites was for the CaCl₂ brine archetype, Type 3. Dr. Hong is correct that the Type 4 archetype samples came from multiple depths at a single site in the Great Australian Bight. The justification for the choice of the Great Australian Bight (large range of depths/alteration, consistency with other well-known carbonate locations) is in the Methods, Section 4.4. We considered moving those paragraphs up. But to avoid getting deep into the weeds so early in the paper, we instead added a note on line 163-164: “The sources and justifications for these archetypes are discussed in more detail in Section 4.4.”

I also have the following minor comments to help improve the readability of the paper.

Done

Line 33-35: this statement also needs to be backed up with references.

Done

Line 36-37: This statement also needs support.

Done

Line 99: GoM needs to be spelled out

Done

Table 2: Why not have the full name for GoM?

Good point, done

Fig.4: To avoid confusion, I suggest change the title of x-axis to “flux due to geothermal convection”, which is also consistent to the title of Fig. 3.

Done

Line 122-124: Can this “predictability” be read from any figure or suppl. Material? Or is it really an important statement that needs to be included?

Deleted

Table 4: Though the calculation for “modified concentration” is not included in the main text, a sentence should still be included to guide the readers to the calculation (such as section 4.4 of the method).

done

Supplementary table 1: Thanks for including detailed calculation here. Why items 4, 5, 9, 10 were indicated as flux, despite the unit is in concentration? Isn’t that you simplify calculate the difference between seawater and the adjusted concentration from the brine?

Good point. We had copied Spencer and Hardie, who maybe should have done better. We changed these to “net chemical contribution” instead.

Line 181, 183, 185: the references 31, 3,1 are easily confused with the exponent. I suggest write “(ref 3)” instead.

done

Line 184: the m from “mc” should be capitalized to be consistent with Eq .1. Refer C_{mor}, Cr, and mc (or Mc) back to table 4 and also include these acronyms in Table 4 to facilitate reading.

Mc corrected, added “(see Table 4 for adjusted water compositions).”

Line 185: is “Q_s” the same as Q_i in Line 132? If yes, use only one of them for consistency. If not, specify the value.

Good catch, thank you. We now use Q_s throughout. (Q_i came from some older calculations that did not make it into the paper.)

Line 197: “but it showed unrealistic level of seawater modifications” I don’t understand this sentence. Which “it” was referred to and why seawater modification? I thought seawater has been corrected for the archetypes?

We clarified this by changing this (now line 201-202) and the following lines to “The pattern of modifications in the theoretical fluid composition needed to balance marine chemical budgets in the absence of low-temperature basalt alteration showed sinks of K⁺, Mg²⁺, and Na⁺ and a source of Ca²⁺. This is consistent with the clastic-derived archetype, but this theoretical fluid showed unrealistic levels of change compared to seawater. Less alteration was required when we used our upper-bound estimate for the groundwater flux, but ...”

Line 198: Again, not quite sure what does seawater modification mean here.

Included in the Line 197 edit

Figure 5: indicated Eq. 1 in the caption to help the readers connect.

done

Eq. 2: is the “mc” in the equation the same as “Mc” in Eq. 1? Is the Q_s the same as Q_i in Line 132? Same comment as for Line 184. Refer back to Table 4 when calling the different terms. What is the value of C_{it} used? I cant see it in Table 4.

Mc and Q_s/Q_i: done, per prior comment about line 184. We deleted this reference to C_{it} because we realized we had already defined it in the prior sentence (which reads, paraphrased: “We calculated the value of C_{it} using Eq 2:”). We deleted the definition of Q_s from line 216 because it was already defined at Eq. 1. We combined the next paragraph with this one to explain more immediately how we calculated C_s.

Line 214: reference 32 can be confused with exponent

done

Figure 6: Where is Eq. 4 indicated in the caption?

Good catch, thanks. It was supposed to be Eq. 2, now corrected.

Line 218: I’m a bit confused here. So C_{it} was being calculated from C_s, which you adopted the

composition of the clastic and CaCl₂ types? This could probably be already indicated in Line 211-214. See our corrections for the comment about Eq. 2. We combined the two paragraphs to make this clearer (see lines 216-225).

Thanks again to Dr. Hong for a really close reading of this paper, which has certainly improved it.

Reviewer #2 (Remarks to the Author):

I have reviewed the revised submission of Wilson et al. and find the revisions to be comprehensive and effective. This is an exciting contribution that ties together a lot of lines of evidence to solve a fundamental question about marine geochemical budgets and hydrogeologic processes. A few very minor remaining suggestions:

- In the new conceptual figure (1A), it would help to label the fluxes (i.e. "QrCr next to 1, etc).

We removed the equation from Fig 1A for purposes of the paper, because it was complicated (and premature, really) to introduce and define all the variables this early. Better to wait for Eq 1.

- Figures 3 and 4 could be combined (perhaps with the compaction-driven fluxes shown in one color above, and the geothermal fluxes in another color below).

This is a good suggestion, but we are, respectfully, terrified to change figure numbers at this stage. We rationalized our cowardice by noting that the sources of errors and hence graphics differ between the two plots, which makes it somewhat complicated to combine them. We hope that reviewers and editors will overlook our cravenness, given that the paper fits within the word and figure limits.

- I could not find a reference to the source of information on lithology (Supplementary Figures 1 and 2). Was it the Baldwin and Butler reference? If so, I'd suggest putting this citation in the captions.

We added this statement to the captions for Supplementary Figures 1 and 2: "Stratigraphy compiled from the sources listed in Table 2." And, thanks again for the great suggestions last time.